Subject Area:
immunology/microbiology/molecular biology

Keywords:
VSG switching, DNA damage, homologous recombination

Author for correspondence:
Lucy Glover
e-mail: lucy.glover@pasteur.fr

# Escaping the immune system by DNA repair and recombination in African trypanosomes

Núria Sima[1], Emilia Jane McLaughlin[1], Sebastian Hutchinson[2]
and Lucy Glover[1]

[1]Trypanosome Molecular Biology, Department of Parasites and Insect Vectors, and [2]Trypanosome Cell Biology and INSERM U1201, Department of Parasites and Insect Vectors, Institut Pasteur, 25–28 Rue du Docteur Roux, 75015 Paris, France

LG, 0000-0001-7191-6890

African trypanosomes escape the mammalian immune response by antigenic variation—the periodic exchange of one surface coat protein, in *Trypanosoma brucei* the variant surface glycoprotein (VSG), for an immunologically distinct one. *VSG* transcription is monoallelic, with only one *VSG* being expressed at a time from a specialized locus, known as an expression site. *VSG* switching is a predominantly recombination-driven process that allows *VSG* sequences to be recombined into the active expression site either replacing the currently active *VSG* or generating a 'new' *VSG* by segmental gene conversion. In this review, we describe what is known about the factors that influence this process, focusing specifically on DNA repair and recombination.

## 1. Introduction

Pathogens have evolved to survive in environments that are often hostile to them. Common to several, including protozoan parasites, bacterial and fungal species, is escape of the host immune response by antigenic variation, the periodic exchange of one antigen for an immunologically distinct one. Although antigenic variation is a common tactic, none have dedicated as much of their genome to this process as *Trypanosoma brucei*. The vector-borne protozoan parasite *T. brucei* is the causative agent of human African trypanosomiasis (HAT) and animal African trypanosomiasis (AAT), and remains today a pervasive public health issue in sub-Saharan Africa.

During an infection, an individual trypanosome will express a single variant surface glycoprotein (VSG) that will be periodically cleared by the host immune system. A small proportion of parasites escape this antibody-mediated clearance by switching the expressed *VSG*; this is a continual process that will go on for as long as the host survives. Indispensable to antigenic variation in trypanosomes is the large family of hypervariable *VSG* genes [1]. A massive expansion of *VSG* genes has resulted in approximately 2000 genes and pseudogenes being dedicated to this gene family. This is between one and two orders of magnitude larger than other pathogens that use antigenic variation as an immune evasion mechanism. This expansion is probably due to selective pressure exerted upon trypanosomes to maintain antigen diversity. Among other protozoan parasites, the variant surface antigen gene families number, for example, approximately 60 *var* genes in *Plasmodium falciparum* [2], approximately 150 *variant surface proteins* (*VSP*) in *Giardia lamblia* [3] and 15 *vsl* genes in *Borrelia burgdorferi* [4]. Between African trypanosome species, which include *T. b. gambiense*, *T. b. rhodesiense*, *T. equiperdum*, *T. congolense* and *T. vivax*, antigenic variation is commonly used but *VSG* gene diversity is dictated by the scale of recombination within each species, at least for *T. b. brucei*, *T. congolense* and *T. vivax* [5].

Trypanosome infections were previously thought to be confined to the blood and cerebrospinal fluid, but parasites have been detected in the skin of asymptomatic patients who were negative for the presence of parasites in the blood [6], and have been shown to reside in the adipose tissue of rodent models [7]. Both HAT and AAT are potentially fatal and treatment of the disease may be further complicated by the tropism of the parasite. Additionally, we do not know the effect of compartmentalization in extravascular spaces on antigenic variation, if any, although the prospect is an intriguing one.

Trypanosomes have evolved into specialists of immune evasion, and several factors in particular facilitate this process: first, approximately 20% of the trypanosome nuclear genome encodes for subtelomeric genes, the majority of which are *VSG* genes [1] that provides a large antigen repertoire. Second, recombination among *VSG* genes further increases the diversity [8–11]. Third, the ability to switch the expressed *VSG* allows the trypanosomes to continuously stay ahead of the immune response. Fourth, strict monoallelic *VSG* gene expression ensures that the immune system is only exposed to a limited number of VSGs at a time and finally, extremely high rates of recycling of the VSG coat ensures that low titres of bound antibodies can be rapidly internalized and destroyed [12,13]. In this review, we summarize what is known about the mechanisms by which trypanosomes undergo antigenic variation and switch the expressed *VSG*, focusing on what is known in *T. b. brucei*. We will describe the *VSG* genomic environment, the mechanisms of VSG switching, DNA double-strand break as a trigger for VSG switching, how chromatin components influence this process and future avenues for study of antigenic variation.

# 2. The *VSG* and its genomic environment

## 2.1. The variant surface glycoprotein

In the bloodstream, where trypanosomes are exposed to the host adaptive immune response, $10^7$ VSG molecules cover the cell surface [14]. VSGs are attached to the cell by a glycosylphosphatidylinositol (GPI) anchor [15], forming a coat, which presumably protects the cell from complement-mediated lysis [16,17] and shields invariant surface molecules also present on the surface. These surface molecules include the transferrin receptor and the haptoglobin–haemoglobin receptors which are required for nutrient uptake [18–20]. The presence of these invariant proteins on the surface poses a problem for the parasite as their antigens do not vary, and so could be cleared by the immune system, but they extend above the VSG monolayer. How the parasite is able to evade destruction in this context is not understood. The rapid rate of antibody endocytosis seen in trypanosomes may facilitate immune evasion [12]. Although this dense protective coat forms the basis for immune evasion mechanism employed by trypanosomes, the VSG coat is itself highly immunogenic [21,22]. At the sequence level, the *VSG* is separated into two distinct domains: the hypervariable N-terminal domain which is 300–350 amino acids long, which is exposed to the immune system, and the conserved C-terminal domain of approximately 40–80 amino acids [9,23], which is buried in the coat [24]. Previously, amino acid sequence diversity alone was thought to be sufficient to sustain long-term antigenic

variation; however, recent findings also implicate post-translational modification (PTM) by the addition of *O*-glycosylation in expanding the immunological space available to this parasite, and therefore increasing the trypanosomes ability to escape the host's adaptive immune system [25].

## 2.2. *VSG* expression site

The trypanosome genome is organized into 11 diploid megabase chromosomes, 5 intermediate chromosomes and approximately 100 minichromosomes [26,27]. *VSG* genes are transcribed from subtelomeric loci called the expression sites (*VSG*-ESs), which are found on the megabase and intermediate chromosomes [28]. Unusually, *VSG* gene transcription is driven by RNA Polymerase I (RNA Pol I) [29], at an extra nucleolar focus termed the expression site body (ESB) [30,31]. *VSG*-ESs have a conserved structure [28,32], they are polycistronic transcription units approximately 60 kb in length and encode for several protein coding genes termed expression site associated genes (*ESAGs*) along with the *VSG*, which is always found immediately adjacent to a telomere (figure 1). There are approximately 15 *VSG*-ESs in the genome and monoallelic *VSG* gene expression dictates that only one *VSG*-ES is active at any one time, while the others are silenced. The *VSG*-ESs and archival *VSG* genes and pseudogenes are found in the subtelomeric regions of the mega/intermediate and mini chromosomes in *T. b. brucei* [26]. It is unclear why silent *VSG* arrays are located in the subtelomeres. This location may aid in the expansion of the *VSG* gene archive via recombination and VSG switching [33]. These regions also fall outside of the Pol II polycistronic transcription units and archival *VSGs* lack promoters—so may additionally prevent illegitimate *VSG* transcription. The importance of nuclear spatial organization for recombination was recently demonstrated by deletion of histone variants H4.V and H3.V in trypanosomes, which revealed a simultaneous increase in *VSG* gene clustering, DNA accessibility across the *VSG*-ES and VSG switching [34]. These data suggest that changes in chromatin structure and spatial organization of the ES might be first steps during recombination events.

The unusual nature of the *VSG*-ES and *VSG* transcription has led researchers to query whether nuclear organization further primes trypanosomes for VSG switching. Disrupting higher-order genome architecture by either loss of a functional cohesin complex [35], which links sister chromatids during replication, or nuclear periphery protein-1 (NUP-1), a lamin-like protein [36], resulted in an elevated level of switching, specifically transcriptional switching in the case of the cohesin complex. The active *VSG*-ES itself occupies a distinct chromatin structure as it is depleted of nucleosomes [37–39]. Additionally, the *VSG*-ES-associated proteins, including RNA Pol I, are sumoylated by an E3 ligase [40]. The high rates of RNA Pol I transcription are facilitated by the deposition of TDP1, a high-mobility group box protein, at the active *VSG*-ES [41]. This more open chromatin structure possibly evolved to allow for the high levels of transcription needed to maintain the volume of *VSG* mRNA required to form an intact coat.

# 3. VSG switching

VSG switching is primarily a recombination-driven process, with *VSG* gene conversion (GC) events that replace the expressed *VSG* with a silent *VSG* gene dominating in

royalsocietypublishing.org/journal/rsob    *Open Biol.* **9**: 190182

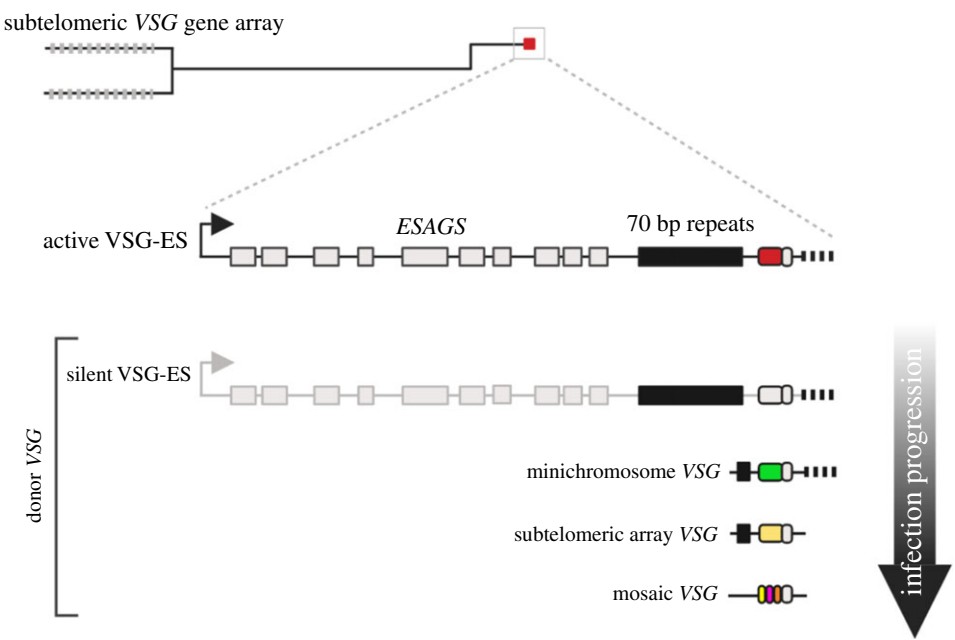

**Figure 1.** Genome architecture and antigenic variation in *T. brucei*. Schematic of a single megabase chromosome with subtelomeric *VSG* arrays (light grey bars). The *VSG*-ES is found proximal to the telomere (red box) and is composed of a single RNA Pol1 promoter approximately 60 kb upstream of the *VSG* gene. Several *expression site associated* genes (ESAGs—light grey boxes) and a single *VSG* gene (red box), adjacent to the telomere, make up this polycistronic locus. During antigenic variation, donor *VSGs* from silent *VSG*-ES, minichromosomes or subtelomeric *VSG* arrays are used. As the infection progresses, mosaic gene formation contributes as well. The 70 bp repeats, *VSG* gene, *VSG* 3'UTR and the telomere provide homology for GC events. 70 bp repeats, black box; *VSG* gene, red, grey, green, yellow or multi-coloured box; *VSG* 3'UTR, small grey box immediately adjacent to the *VSG* gene; telomere, black bar. Figures created with Biorender.com.

trypanosomes [42] (figure 1). VSG switching occurs independently of the host immune response [43,44]. Less frequently, VSG switching occurs via *in situ* switching, with the silencing of the active ES and activation of a silent ES, without any DNA rearrangements. These recombination reactions are driven by homology found both in the *VSG*-ES and the genomic archive, and include the blocks of 70 bp repeats which are found upstream of most archival VSGs, and highly conserved elements within the *VSG* 3'-untranslated region (UTR) which serve as recombination substrates for repair and subsequent VSG switching [1,9,10,45–48]. The 70 bp repeats mark the 5' boundary of GC events, serving as 'recombination hotspots' [1,9,49], promoting access to the archival *VSGs* [46], thereby increasing the VSG switching diversity [47,50]. A large proportion of the *VSG* archive is composed of pseudo-*VSG* genes, which may act as an information pool for the formation of mosaic *VSG* genes. The generation of mosaic *VSGs* involves intragenic segmental conversions and predominate in a late infection [8,11,51] (figure 1). The expressed *VSG* diversity in a trypanosome infection was underappreciated until work by Hall *et al.* [8] and Mugnier *et al.* [11] revealed the true scale of antigenic variation. Up to this point, the antigenic variation paradigm was shaped by an understanding that the waves of parasitaemia seen in patients were composed of a homologous or limited number of *VSG*-expressing parasites. However, the number of distinct VSGs detected in each population may be as high as 80, but is probably higher still [8,11]. This is the equivalent to approximately 5% of the silent *VSG* repertoire at any one time. This suggests that, for trypanosomes, the time to transmission should be less than the time to *VSG* repertoire exhaustion. Within these diverse populations, mosaic *VSG* genes are predominantly seen later in an infection and are most likely to be formed just prior to

expression [8,9,11] (figure 1). Replacing the VSG coat was also recently shown to be a slower process than expected. It takes approximately 4.5 days to replace the entire coat, but the VSG being replaced is only recognizable to host antibodies for the first day [52].

It remains unclear what specifically leads to a VSG switch event. Antigenic variation is probably a stochastic process; however, telomere instability, triggered by deletion of telomerase, the enzyme responsible for maintaining telomere length or knockdown of proteins important for telomere integrity, leads to higher switching frequencies [50,53–57]. The telomere-associated proteins TbRAP1 (Repressor activator protein 1 [58]) and TbTRF (TTAGGG repeat factor [56]) along with TbTIF2 (TRF1 interacting factor 2 [56]) appear to suppress ES recombination events by maintaining subtelomeric chromatin integrity. The depletion of these proteins leads to an accumulation of DNA damage at the *VSG*-ES and increased switching [56,58] (figure 2). Components of the inositol phosphate pathway, here phosphatidylinositol 5-kinase (TbPIP5 K) and phosphatidylinositol 5-phosphatase (TbPIP5Pase), modulate switch frequency through interactions with telomeres [59]. Indeed, TbPIP5Pase has been shown to interact with TbRAP1 [59,60]. Recently, several proteins involved in DNA repair or genome integrity, such as the replication factor A (RPA), the translesion polymerases PrimPol-like 2 (PPL2) [61] and DNA Polymerase Nu (PolN), were isolated from telomere pull-downs [62], suggesting repair proteins accumulate at telomeres, perhaps to ensure any damage is rapidly repaired so as to preserve telomere integrity. The histone methyltransferase DOT1B (disrupter of telomeric silencing) is not associated with telomere stability, but appears to regulate *in situ* switching kinetics [63].

How do we determine switching frequencies in trypanosomes? There are various methods, but the most commonly

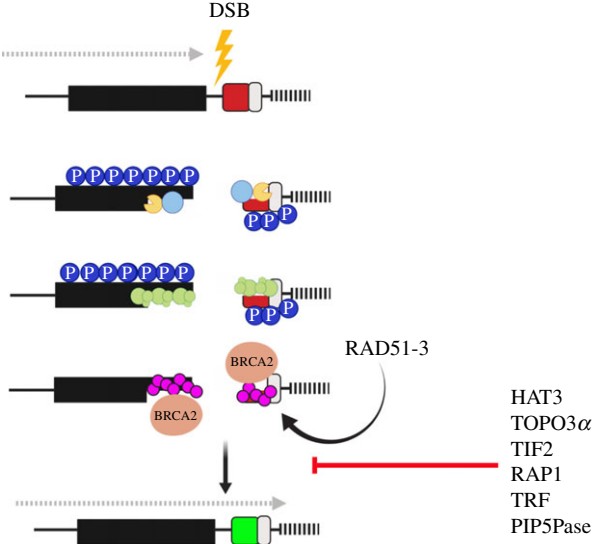

**Figure 2.** DNA repair in *T. brucei*. Following a double-strand break (yellow lightning bolt) at the active VSG-ES, Histone H2A is phosphorylated (dark blue circle with P), and the MRN complex (light blue circle) binds and initiated resection by recruiting endo and exonucleases (yellow pacman). Following resection, the RPA complex (green circles) binds to the ssDNA. RPA is displaced by RAD51 (pink circle), which is loaded onto the ssDNA by BRCA2 (peach oval) facilitated by RAD51-3. Resolution of the break results in VSG switching. Direction of transcription, dotted grey arrow; 70 bp repeats, black box; telomere, black bars; VSG switching suppression, red line. Figures created with Biorender.com.

used are either selection by resistance marker or assessment of total switching frequency. *VSG*-ES can be tagged with either positive or negative selectable markers, and the frequency of parasite survival analysed following antibiotic selection. Using positive selectable markers, *in situ* switching frequencies are estimated at between $1 \times 10^{-4}$ and $6 \times 10^{-7}$, depending on the *VSG*-ES [64,65], and are similar for negative selectable marker, at approximately $10^{-5}$ [66]. Using immune selection, Lamont *et al.* [67] infected immunized mice and selected for switched variants derived *in vitro*, calculating the switching frequency between $2.2 \times 10^{-7}$ and $2.6 \times 10^{-6}$ switches per cell division, whereas Davies *et al.* [68] used *in vitro* antibody selection, calculating background-switching rates below $1 \times 10^{-6}$. Alternatively, basal switching frequencies have been assessed using magnet-activated cell sorting, which estimates switching frequencies to be between $5.9 \times 10^{-6}$ and $0.85 \times 10^{-5}$ [46,50,56,69–71]. In trypanosomes, because *VSG* expression is essential, RNAi against the active *VSG* transcript results in cells that have switched to *VSGs* that are not targeted by RNAi. The switching frequency following *VSG* RNAi is estimated to be at a rate of $10^{-4}$ per division, predominantly by *in situ* switching [72]. There is some evidence that laboratory adaptation has reduced the switching frequency displayed by trypanosomes, as there is evidence that completion of the parasite life cycle is important to reset the VSG switching frequency: direct comparison of syringe-passaged to tsetse fly-passaged lines revealed VSG switching frequencies of approximately $10^{-6}$ and approximately $10^{-2}$, respectively [73]. While these data are instructive, they rely on experimental systems which are far from ideal. The strains used are largely culture-adapted monomorphic strains, which seem to display greatly reduced switching frequencies. The experimental

limitations are, however, understandable, given the difficulty of the task. The ideal situation would be to measure the switching frequency in a chronic model infection; however, this is not possible using current technology. The predominant form of switching in trypanosomes is via DNA recombination, yet the majority of the above-mentioned techniques measure *in situ* switching frequencies, or indeed, in the case of *VSG* RNAi, result in *in situ* switchers. It remains to be seen whether the current data on VSG switching frequencies accurately reflect long-term trypanosome infections.

## 4. DNA repair, recombination and VSG switching

Despite intense study, we still do not fully understand what leads to VSG switching in a natural context. Cleavage within the 70 bp repeats by a specific endonuclease has been hypothesized [74], similar to that seen in yeast mating type switching [75], but is unlikely given the trypanosomes' ability to undergo GC in the absence of the 70 bp repeats [47], and no such enzyme has yet been identified. We have determined that subtelomeric regions in *T. b. brucei* are fragile [69,76], and naturally occurring DSBs form in a transcription-independent manner [76]. These DSBs may be formed due to instability within the AT-rich 70 bp repeats, rendering the region prone to DNA damage [69]. Indeed, the AT repeats form an unstable non-H-bonded structure in plasmid DNA [77]. Alternatively, high levels of transcription at the *VSG*-ES may result in either collapse of the replication fork or clashes between the replication and transcription machinery, both of which could result in DSBs [76,78,79]. However, as silent *VSG*-ESs are also fragile, it is unlikely that high levels of transcription alone drive the formation of DNA breaks. R-loops, or RNA-DNA hybrids, at the active *VSG*-ES lead to an accumulation of DNA damage [80]. Although natural DSBs have been detected at the *VSG*-ESs, the effect of a DSB on VSG switching is more nuanced. The position of the DSB is important in predicting whether a break leads to a VSG switch or is repaired without inducing antigenic variation, and this is dictated by the double-strand break response (DSBR) pathway and the mechanism of switching [76]. Only DSBs at the active *VSG*-ES lead to VSG switching, and the most productive are those breaks formed between the *VSG* gene and the 70 bp repeats [69,76]. There is also a clear hierarchy in the donor *VSG* selection for recombination and switching [72]. Telomeric *VSGs* from *VSG*-ESs and minichromosomes are the predominant donors in the early stages of infection, with array *VSGs* and mosaic *VSGs* observed later in a trypanosome infection [8,11] (figure 1). It is unclear from the available data whether this hierarchy is driven purely by trypanosome factors or is a product of immune memory.

The homologous recombination (HR) pathway is largely conserved in *T. b. brucei*, but some components show significant sequence diversity, which suggests there may be functional divergence within the pathway [81]. This diversity may have evolved from the dependence on recombination pathways for antigenic variation. Classically, DNA damage elicits a $G_2/M$ checkpoint that prevents division of unrepaired DNA into the daughter cells, and so preserves genome integrity. Trypanosomes appear to forgo this

royalsocietypublishing.org/journal/rsob   Open Biol. **9**: 190182

checkpoint and continue to replicate and divide their DNA, which suggests they are able to tolerate DNA damage to a greater extent than other eukaryotes [54,82]. The advantage may be to allow for greater time for homology searching and antigenic variation, at least for a break at a *VSG*-ES. HR dominates as the major form of RAD51-dependent repair, with microhomology-mediated end joining (MMEJ) acting as the RAD51-independent alternative repair pathway—but both are important for VSG switching [69,76,83]. Non-homologous end joining appears to be absent in trypanosomes [84], while HR is conserved in *T. brucei*, and several components show a detectable role in antigenic variation. The early sensor in the DSBR is the MRN complex (MRE11, RAD50, NBS1 in mammals/XRS2 in *Saccharomyces cerevisiae*) and it regulates both HR and MMEJ. MRE11 licences the repair by initiating resection which is important for strand invasion [85] (figure 2). In *T. brucei* and *Leishmania*, MRE11 has a critical role in maintaining genomic integrity and null mutants are hypersensitive to DNA-damaging agents, but interestingly appears to be dispensable for VSG switching [86–89], suggesting another exonuclease initiates resection, or is able to complement MRE11 loss. What then drives the DNA damage response (DDR) during antigenic variation? Two RecQ-like helicases have been identified in trypanosomes and *recq2* mutants show elevated *VSG* switching by telomere recombination in addition to *VSG* GC events [78]. As *VSG* expression is essential [72,90], these observations suggest that *T. brucei* is able to invoke multiple repair pathways to ensure VSG switching.

RAD51, the primary recombinase in DNA repair, is required for homology searching and DNA strand exchange, and is loaded onto single-strand DNA by breast cancer gene 2 (BRCA2), via an expanded number of BRC repeat motifs, displacing RPA [91,92] (figure 2). In *T. brucei*, BRCA2 is essential for HR, DNA replication, cell division and antigenic variation [93,94]. RAD51 is essential for HR, and null mutant cells display cell growth retardation and an increased sensitivity to DNA-damaging agents [95]. In its absence, MMEJ is the primary repair pathway [96,97]. Consistently, RAD51-deficient parasites have impaired, but not abolished, VSG switching [71,95,96,98], again revealing how trypanosomes will use alternative repair pathways to ensure VSG switching. Indeed, only 60% of repair at a *VSG*-ES is RAD51-dependent, suggesting less stringent requirements for repair pathway choice, which may allow for greater antigenic variation [76]. Five RAD51-related proteins (RAD51-3, 4, 5, 6, and DMC1 [99]) have been identified, and all have roles in DSBR and are essential for the subnuclear localization of RAD51 in response to damage. In particular, RAD51-3 also contributes to VSG switching [99] (figure 2). Recombination-dependent VSG switching occurs mainly by GC events, where the active *VSG* is deleted and replaced by the duplicative copying of a silent donor [42,44,100,101]. In addition, crossover switching events, where two *VSGs* are exchanged, have been observed [72,101–103]. The RTR complex, which includes the RecQ-family helicase, a Topoisomerase III $\alpha$ and RMI1/2, suppresses mitotic crossover and removes recombination intermediates [104]. In trypanosomes, components of the RTR complex act in two pathways that lead to DNA repair-linked VSG switching. In the first, *Tb*TOPO3$\alpha$ and *Tb*RMI1 suppress *VSG* GC and VSG crossover events (figure 2), whereas the second is dependent on RAD51 and RMI1 [71,105]. Moreover, in the absence of *Tb*TOPO3$\alpha$, active *VSG*-ES recombination events

predominantly use ESAGs and not the 70 bp repeats as a recombinational substrate [71].

We are now discovering that HR requires both genetic factors and PTMs of histones and non-histone proteins for the initiation and execution of the repair response [106,107]. These marks affect chromatin condensation and serve as recognition sites, promoting binding of repair factors [107]. Indeed, a cycle of acetylation–deacetylation has been proposed that promotes a more open chromatin state, facilitating the recruitment of repair factors and subsequently the restoration of the chromatin architecture following the repair [108]. In *T. brucei*, the histone acetyltransferase HAT3, which modifies histone H4K4 [109], and the histone deactylase SIR2rp1 [110] are required for HR repair and RAD51 foci assembly and disassembly, respectively [98]. HAT3's function in DNA repair was shown to be locus-specific and impact VSG switching; at a chromosome internal locus, HAT3 promoted DNA resection and RAD51 focal assembly but suppressed resection at a *VSG*-ES. The consequence of this is that in the presence of HAT3 *VSG* GC events, which required resection to expose the 70 bp repeats and provide the homology required for repair, are suppressed [98] (figure 2). This suggests that specific chromatin marks may regulate DNA recombination and VSG switching. Additionally, phosphorylation of H2A(X) is central to triggering the protein cascade required to initiate DSBR [111]. In *T. brucei*, H2A Thr130 is phosphorylated in response to a DSB and colocalizes with RAD51 and RPA repair foci, typically during S or G2 phases of the cell cycle [82,112] (figure 2).

# 5. Future avenues for study of antigenic variation

It is only in the past few years that we have begun to understand the depth and extent of trypanosome antigenic variation. The simplistic view of the arms race between trypanosomes and the host immune response, with trypanosome populations constantly remaining one step ahead, must now consider that each wave of parasitaemia is a mix of distinct *VSGs* and mosaic *VSGs*, and each potentially with PTMs that further separates them from each other.

There are, however, outstanding questions that remain to be answered:

— DNA breaks can act as potent triggers for antigenic variation, but what governs the specific DSBR that determines the repair pathway choice, and how does the resolution of a DNA break result in recombination events that are able to generate mosaic *VSG* genes?
— Studies on DNA breaks in the active *VSG*-ES have thus far employed the I-*Sce*I meganuclease system, where both RAD51-dependent and -independent break-induced repair are used. Given that there is a loose hierarchy of VSG switching, does repair pathway choice change over time and, if so, how is it regulated?
— What does the trypanosomes gain by colonizing the skin and adipose tissue and are different cohorts of *VSG* genes expressed in these regions? (See [113] for more detail on tissue tropism.)
— What is the full complement of PTMs on VSGs, and how does this additional layer of antigenicity expand the parasites ability to escape the immune system?

— Finally, studies in antigenic variation have been mostly been confined to *T. b. brucei* due to their genetic tractability. Whether antigenic variation operates in a similar manner and what drives the process in *T. congolense* or *T. vivax* is virtually unknown.

These are just a few of the many open questions facing researchers, which makes this, still, a fascinating area of study.

Data accessibility. This article has no additional data.

Authors' contributions. N.S., E.J.M., S.H. and L.G. wrote the paper.

Competing interests. The authors declare we have no competing interests.

Funding. L.G. is supported by ANR JCJ (VSGREG; grant no. ANR-17-CE12-0012) grant and Institut Pasteur G5 position. N.S. is supported by an Institut Pasteur post-doc position. E.J.M. is a PPU doctoral student; this project has received funding from the European Union's Horizon 2020 research and innovation programme under the Marie Sklodowska-Curie grant agreement no. 665807 and is registered a student at Université Paris Descartes, Paris, France. S.H. is a Marie Curie fellow; this project has received funding from the European Union's Horizon 2020 research and innovation programme under the Marie Skłodowska-Curie grant agreement no. 794979.

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
