## [Reviewer comments · Open Biology]

Review History

RSOB-19-0182.R0 (Original submission)

Review form: Reviewer 1

Recommendation

Accept with minor revision (please list in comments)

Do you have any ethical concerns with this paper?

No

Comments to the Author

Teruel et al provide a timely review of the importance of DNA repair and recombination to antigenic variation in *Trypanosoma brucei*. Their review complements recent reviews by Aresta-Branco et al (parallels with antibody diversification; PMID: 30826207), da Silva et al (mechanisms of DNA lesion generation; PMID: 30440029), Bangs (evolution of antigenic variation; PMID: 30370931), and Cestari and Stuart (transcriptional regulation; PMID: 29491740).

The review is well organised, clearly presented and addresses important issues of relevance to those interested in antigenic variation, as well as the wider area of DNA repair. However, there

are a few issues that should be addressed prior to publication, as well as a number of typos (see list after comments).

Major comments

1. The authors provide a detailed summary of switch rate variation seen between strains and using different approaches (lines 190-208). How might these variations influence or complicate the authors' analyses and those of others in the field? Is there a risk that the culture adaptation effect (line 207) biases the types of DNA repair (and breaks) that drive antigenic variation? Is this issue relevant to investigation of the apparent switching mechanism hierarchy (line 166, 235, 324, 328)?
2. The description of the role of HAT3 in repair and VSG switching is confusing (lines 298-303) – as written, the second sentence doesn't logically follow the first. The authors should include more detail, emphasising the different effects of HAT3 at internal chromosomal loci versus VSG expression sites; hence HAT3's suppressive effect on VSG switching (Glover & Horn 2014; PMID: 25300492).

Minor comments

3. Some of the word choices have unintended implications. For example, 'cyclical' (line 49) suggests a return to a previous state, whereas antigenic switching and clearance of specific antigenic types by the adaptive immune system results in the progressive expansion and clearance of new variants, i.e. a process of ongoing change enabling persistence. Also, 'several orders of magnitude' (line 53) seems an exaggeration, given the numbers provided at lines 56-8.
4. I noted some inconsistencies, e.g. 'one third' (line 72) contradicts the text at line 52 (10%), and ESAGs don't directly contribute to the variant antigen repertoire as this sentence implies. The statement at line 259-61 seems inconsistent with that at line 143-4 – 'switching primarily driven by gene conversion' (143), 'recq2 mutants show elevated switching by [...] gene conversion. [...] a marked shift in the type of switching used [...] compared to wild type cells'. The authors should also review for repetitive text (e.g. at lines 283-4).
5. Rather than using the term 'increasing virulence' (line 105), the authors should be more specific and say that the addition of O-glycosylation to the already extensive VSG variability potentially extends the ability of the parasite to evade the host's adaptive immune system. The authors should also be more precise in their description of the position of the most productive DSBs in the active VSG-ES (line 233). My understanding is that this is thought to be limited to an upstream region proximal to the VSG, probably in or close to the 70 bp repeats (rather than further upstream in the ESAGs or towards the distal upstream ES promoter).
6. The description of the RNAi experiments, the emergence of switch variants, and the impact of culture adaptation on switch frequency (lines 202-8) needs rewriting for clarity. In particular, the relevance of the essential nature of VSG in this context is unclear and it should be stated that the actively expressed VSG is specifically targeted, enabling a switch to a new, non-targeted variant. Also, I was unsure as to the relevance of VSG-associated growth rates to this discussion (lines 208-11).
7. The numbers of mini and intermediate chromosomes are less constrained than stated at line 108. There are ~100 minichromosomes and intermediate chromosome ploidy varies between strains (see Wickstead et al 2004; PMID: 15173109).
8. At line 276, the authors state there are five RAD51-related proteins but list four, omitting DMC1 (Proudfoot and McCulloch 2005; PMID: 16326865).

9. As the focus of the review is DNA repair, I suggest the authors re-order their final questions to give primacy to those pertinent to this topic in their list starting at line 317.

Typos and suggested edits

Line 56 ('...families number, for example, ~60 var genes'); 111 (state that VSG expression sites are found at the sub-telomeres of megabase chromosomes); 121 (delete 'they'); 137 (delete second 'is facilitated by'); 131, 134 ('cohesin'); 150 ('untranslated'); 177 ('suppress'); 180-1 (put brackets around abbreviations); 186 ('preserve'); 191 ('uses', 'assesses'); 194 ('in situ switching frequencies are estimated at'); 216 ('seen in yeast', 'given'); 242 ('preserves'); 254 (italicise 'Leishmania'); 255 ('hypersensitive to DNA'); 259 & 301 (consistent terminology for null mutants); 261 (delete 'that'); 267 ('via an expanded'); 271 ('DNA damaging agents'); 278 ('RAD51 in response to DNA damage'); 283 ('suppresses'); 294 (delete 'a'); 317 ('does the trypanosome gain by'; 'VSG genes expressed in these'); 321 ('antigenicity expand the parasite's ability'); 328 ('Given that there is a loose hierarchy of VSG switching, does repair pathway choice change over time and, if so, how is this regulated?'); 357 ('single megabase chromosome'); 358 ('found proximal to the telomere'); 360 ('light grey boxes'); 361 ('variation, donor VSGs from'); 364 (delete 'highlighted in black'); 371 ('MRN complex (light blue circle) binds and initiates resection').

Review form: Reviewer 2 (Igor Cestari)

Recommendation

Accept with minor revision (please list in comments)

Do you have any ethical concerns with this paper?

No

Comments to the Author

This is a helpful review by Teruel and colleagues. The work submitted here provides a good overview with some discussions (not detailed) on the mechanisms underlying antigenic variation in African trypanosomes focusing on recombination. The work covers well the literature and discusses many possible mechanisms revolving VSG recombination and potential mechanisms that initiate the switching process. It would be nice to have this potential mechanism that initiates the switching numbered to highlight this part of the discussion since a non-expert may not grasp the relevance of the content here.

There are minor faults on the content which can be easily fixed, and I have provided some comments that will help the authors. This manuscript has a vast number of grammatical mistakes, some of which is pointed out here (but not all). Hence, I suggest the authors review their manuscript carefully.

Detailed comments:

line 51 - A massive expansion in VSG genes have resulted in ... Is this meant to be a massive expansion of VSG genes that resulted in ...?

line 72: the sentence " and their associated sequences..." is unclear. Is this meant to be VSG related sequences or VSG pseudogenes

line 82: "as a trigger", it is better to complete the sentence: a trigger for switching if that is what you mean.

Line 88: " VSGs are attached to the cell by a glycosylphosphatidylinositol (GPI) anchor (14), which protects the cell from complement mediated complement mediated lysis (15) and shields invariant surface molecules also present on the surface." This sentence is unclear because it indicates that GPIs confer resistance to complement. GPIs anchors the VSG protein on the surface. Also, the reference 15 (Sheader K et al. PNAS 2005) do not show the involvement of VSGs and complement resistance. It shows that knockdown of VSG gene results in clearance of the parasite during infection of mouse. The role of VSG in complement resistance was surprisingly superficially studied in this field and here some helpful references that the author can start his/her search: Engstler M et al. Cell 2007; Russo DC et al 1994 Parasitol Res. This is not to say that complement will not have a role, but it appears that other mechanisms of clearance are also involved (e.g., J A Macaskill et al Immunol 1980; Cheung JL et al 2016 Plos Pathogen).

Line 94. Suggested "cleared by the immune system".

Line 95. Extra space before period.

Line 96. Finish the sentence, instead of "this" use "immune evasion" or "immune clearance" etc.

Line 109. I found the sentence convoluted: "VSG genes are transcribed by RNA Polymerase I at an extra nucleolar transcription factory termed the expression site body (ESB)(23, 24) from a subtelomeric locus, called the expression site (VSG-ES)." I also suggest replacing "transcription factory" to "extranucleolar foci" since the function of ESB remains unclear. Rewrite the sentence, please.

Line 112. The reference 25 is wrong. I suggest Becker M et al. 2004 Genome Res.

Line 113. The sentence seems wrong: "and encode of several proteins coding genes". Check English.

Line 137: "high rates of Pol 1 transcription are facilitated by is facilitated by the deposition of TDP1" Check duplication of words.

Line 361: Fig legend: "During antigenic variation donor VSGs are from a silent VSG-ES, minichromosome, or subtelomeric VSG array are used." Check the structure of this sentence.

Line 164. "This suggests that trypanosomes must ensure that the time to transmission is less than the time to VSG repertoire exhaustion." It may be better to rewrite this sentence as trypanosomes do not "ensure". Transmission may occur before VSG repertoire exhaust, and exhaustion of VSG gene repertoire may not occur due to recombination potential.

Line 179. "Components of the inositol phosphatase pathway, here phosphatidylinositol 5-kinase - TbPIP5Pase and phosphatidylinositol 5-phosphatase - TbPIP5K, modulate switch frequency through interactions with telomeres (53)." It should be inositol phosphate pathway instead of inositol phosphatase pathway. Also, the abbreviations for PIP5K and PIP5Pase seems inverted. PIP5K is the kinase and PIP5Pase is the phosphatase.

Line 181. "Indeed, TbPIP5Pase has been shown to interact with TbRAP1 and TbTRF (52)." The reference provided is wrong (52, Jehi SE et al. NAR 2014). Jehi did not show interactions of PIP5Pase with RAP1. The correct reference should be Cestari and Stuart PNAS 2015 and Cestari et. al 2019 Mol Cell Biol. Also, only RAP1 was found to interact with PIP5Pase, not TRF. This was detailed in Cestari et al. 2019 Mol Biol Cell.

Line 191. "most common used either". It seems that a verb is needed in this sentence.

Line 192. "Selectable markers enable VSG-ES to be tagged with either positive or negative selectable markers". The sentence is cumbersome; please review it. Also, explain better the process. Line 194 says "with drug treatment," which drug, and how does it work?

Line 203. "results cells that have switched ins order to survive" Misspelled words.

Line 216. "similar to that seen yeast mating type switching (69), but is unlikely give the trypanosomes ability to undergo gene conversion in the absence of the 70bp repeats". There are several small grammatical issues in this text which causes a tremendous distraction. From now on I will stop commenting on them and strongly recommend the authors to review the English.

Line 234-235. Please clarify in the text the hierarchy in VSG switching. Does the order or preference in ESs and minichromosomes for VSG switching the consequences of preferred DNA sequences in these sites that may facilitate recombination? If so, are they not appearing so often at later stages of infection because the host has developed antibodies for VSGs in these more favourable recombination sites?

Line 245. I suggest using "the advantage" instead of "The purpose", because this is the result of an evolutionary process.

Line 267. BRCA2 and BRC repeats. Please, state what they stand for. Same to other abbreviations.

Line 269. Reference 69 is a work in yeast and not in trypanosomes. The text mentions the role of TbTOPO3a in ES recombination using 70 bp repeats. Please correct the reference.

Line 292. "The requirements for HR go beyond the mere genetic factors, we are now discovering that" confusing sentence with grammatical mistakes.

Line 318. Review your grammar. Do this to all the text.

Decision letter (RSOB-19-0182.R0)

07-Oct-2019

Dear Dr Glover

We are pleased to inform you that your manuscript RSOB-19-0182 entitled "Escaping the immune system by DNA repair, recombination in African trypanosomes" has been accepted by the Editor for publication in Open Biology. The reviewer(s) have recommended publication, but also suggest some minor revisions to your manuscript. Therefore, we invite you to respond to the reviewer(s)' comments and revise your manuscript.

Please submit the revised version of your manuscript within 7 days. If you do not think you will be able to meet this date please let us know immediately and we can extend this deadline for you.

To revise your manuscript, log into <https://mc.manuscriptcentral.com/rsob> and enter your Author Centre, where you will find your manuscript title listed under "Manuscripts with

Decisions." Under "Actions," click on "Create a Revision." Your manuscript number has been appended to denote a revision.

- 1) A text file of the manuscript (doc, txt, rtf or tex), including the references, tables (including captions) and figure captions. Please remove any tracked changes from the text before submission. PDF files are not an accepted format for the "Main Document".
- 2) A separate electronic file of each figure (tiff, EPS or print-quality PDF preferred). The format should be produced directly from original creation package, or original software format. Please note that PowerPoint files are not accepted.
- 3) Electronic supplementary material: this should be contained in a separate file from the main text and meet our ESM criteria (see <http://royalsocietypublishing.org/instructions-authors#question5>). All supplementary materials accompanying an accepted article will be treated as in their final form. They will be published alongside the paper on the journal website and posted on the online figshare repository. Files on figshare will be made available approximately one week before the accompanying article so that the supplementary material can be attributed a unique DOI.

Online supplementary material will also carry the title and description provided during submission, so please ensure these are accurate and informative. Note that the Royal Society will not edit or typeset supplementary material and it will be hosted as provided. Please ensure that the supplementary material includes the paper details (authors, title, journal name, article DOI). Your article DOI will be 10.1098/rsob.2016[last 4 digits of e.g. 10.1098/rsob.20160049].

- 4) A media summary: a short non-technical summary (up to 100 words) of the key findings/importance of your manuscript. Please try to write in simple English, avoid jargon, explain the importance of the topic, outline the main implications and describe why this topic is newsworthy.

Data-Sharing

It is a condition of publication that data supporting your paper are made available. Data should be made available either in the electronic supplementary material or through an appropriate repository. Details of how to access data should be included in your paper. Please see <http://royalsocietypublishing.org/site/authors/policy.xhtml#question6> for more details.

Data accessibility section

Sincerely,
The Open Biology Team
mailto:openbiology@royalsociety.org

Reviewer(s)' Comments to Author:

Referee: 1

Comments to the Author(s)

Teruel et al provide a timely review of the importance of DNA repair and recombination to antigenic variation in *Trypanosoma brucei*. Their review complements recent reviews by Aresta-Branco et al (parallels with antibody diversification; PMID: 30826207), da Silva et al (mechanisms of DNA lesion generation; PMID: 30440029), Bangs (evolution of antigenic variation; PMID: 30370931), and Cestari and Stuart (transcriptional regulation; PMID: 29491740).

The review is well organised, clearly presented and addresses important issues of relevance to those interested in antigenic variation, as well as the wider area of DNA repair. However, there are a few issues that should be addressed prior to publication, as well as a number of typos (see list after comments).

Major comments

1. The authors provide a detailed summary of switch rate variation seen between strains and using different approaches (lines 190-208). How might these variations influence or complicate the authors' analyses and those of others in the field? Is there a risk that the culture adaptation effect (line 207) biases the types of DNA repair (and breaks) that drive antigenic variation? Is this issue relevant to investigation of the apparent switching mechanism hierarchy (line 166, 235, 324, 328)?
2. The description of the role of HAT3 in repair and VSG switching is confusing (lines 298-303) – as written, the second sentence doesn't logically follow the first. The authors should include more detail, emphasising the different effects of HAT3 at internal chromosomal loci versus VSG expression sites; hence HAT3's suppressive effect on VSG switching (Glover & Horn 2014; PMID: 25300492).

Minor comments

3. Some of the word choices have unintended implications. For example, 'cyclical' (line 49) suggests a return to a previous state, whereas antigenic switching and clearance of specific antigenic types by the adaptive immune system results in the progressive expansion and clearance of new variants, i.e. a process of ongoing change enabling persistence. Also, 'several orders of magnitude' (line 53) seems an exaggeration, given the numbers provided at lines 56-8.
4. I noted some inconsistencies, e.g. 'one third' (line 72) contradicts the text at line 52 (10%), and ESAGs don't directly contribute to the variant antigen repertoire as this sentence implies. The statement at line 259-61 seems inconsistent with that at line 143-4 – 'switching primarily driven

by gene conversion' (143), 'recq2 mutants show elevated switching by [...] gene conversion. [...] a marked shift in the type of switching used [...] compared to wild type cells'). The authors should also review for repetitive text (e.g. at lines 283-4).

5. Rather than using the term 'increasing virulence' (line 105), the authors should be more specific and say that the addition of O-glycosylation to the already extensive VSG variability potentially extends the ability of the parasite to evade the host's adaptive immune system. The authors should also be more precise in their description of the position of the most productive DSBs in the active VSG-ES (line 233). My understanding is that this is thought to be limited to an upstream region proximal to the VSG, probably in or close to the 70 bp repeats (rather than further upstream in the ESAGs or towards the distal upstream ES promoter).

6. The description of the RNAi experiments, the emergence of switch variants, and the impact of culture adaptation on switch frequency (lines 202-8) needs rewriting for clarity. In particular, the relevance of the essential nature of VSG in this context is unclear and it should be stated that the actively expressed VSG is specifically targeted, enabling a switch to a new, non-targeted variant. Also, I was unsure as to the relevance of VSG-associated growth rates to this discussion (lines 208-11).

7. The numbers of mini and intermediate chromosomes are less constrained than stated at line 108. There are ~100 minichromosomes and intermediate chromosome ploidy varies between strains (see Wickstead et al 2004; PMID: 15173109).

8. At line 276, the authors state there are five RAD51-related proteins but list four, omitting DMC1 (Proudfoot and McCulloch 2005; PMID: 16326865).

9. As the focus of the review is DNA repair, I suggest the authors re-order their final questions to give primacy to those pertinent to this topic in their list starting at line 317.

Typos and suggested edits

Line 56 ('...families number, for example, ~60 var genes'); 111 (state that VSG expression sites are found at the sub-telomeres of megabase chromosomes); 121 (delete 'they'); 137 (delete second 'is facilitated by'); 131, 134 ('cohesin'); 150 ('untranslated'); 177 ('suppress'); 180-1 (put brackets around abbreviations); 186 ('preserve'); 191 ('uses', 'assesses'); 194 ('in situ switching frequencies are estimated at'); 216 ('seen in yeast', 'given'); 242 ('preserves'); 254 (italicise 'Leishmania'); 255 ('hypersensitive to DNA'); 259 & 301 (consistent terminology for null mutants); 261 (delete 'that'); 267 ('via an expanded'); 271 ('DNA damaging agents'); 278 ('RAD51 in response to DNA damage'); 283 ('suppresses'); 294 (delete 'a'); 317 ('does the trypanosome gain by'; 'VSG genes expressed in these'); 321 ('antigenicity expand the parasite's ability'); 328 ('Given that there is a loose hierarchy of VSG switching, does repair pathway choice change over time and, if so, how is this regulated?'); 357 ('single megabase chromosome'); 358 ('found proximal to the telomere'); 360 ('light grey boxes'); 361 ('variation, donor VSGs from'); 364 (delete 'highlighted in black'); 371 ('MRN complex (light blue circle) binds and initiates resection').

Referee: 2

Comments to the Author(s)

This is a helpful review by Teruel and colleagues. The work submitted here provides a good overview with some discussions (not detailed) on the mechanisms underlying antigenic variation in African trypanosomes focusing on recombination. The work covers well the literature and discusses many possible mechanisms revolving VSG recombination and potential mechanisms

that initiate the switching process. It would be nice to have this potential mechanism that initiates the switching numbered to highlight this part of the discussion since a non-expert may not grasp the relevance of the content here.

There are minor faults on the content which can be easily fixed, and I have provided some comments that will help the authors. This manuscript has a vast number of grammatical mistakes, some of which is pointed out here (but not all). Hence, I suggest the authors review their manuscript carefully.

Detailed comments:

line 51 - A massive expansion in VSG genes have resulted in ... Is this meant to be a massive expansion of VSG genes that resulted in ...?

line 72: the sentence " and their associated sequences..." is unclear. Is this meant to be VSG related sequences or VSG pseudogenes

line 82: "as a trigger", it is better to complete the sentence: a trigger for switching if that is what you mean.

Line 88: " VSGs are attached to the cell by a glycosylphosphatidylinositol (GPI) anchor (14), which protects the cell from complement mediated complement mediated lysis (15) and shields invariant surface molecules also present on the surface." This sentence is unclear because it indicates that GPIs confer resistance to complement. GPIs anchors the VSG protein on the surface. Also, the reference 15 (Shedden K et al. PNAS 2005) do not show the involvement of VSGs and complement resistance. It shows that knockdown of VSG gene results in clearance of the parasite during infection of mouse. The role of VSG in complement resistance was surprisingly superficially studied in this field and here some helpful references that the author can start his/her search: Engstler M et al. Cell 2007; Russo DC et al 1994 Parasitol Res. This is not to say that complement will not have a role, but it appears that other mechanisms of clearance are also involved (e.g., J A Macaskill et al Immunol 1980; Cheung JL et al 2016 Plos Pathogen).

Line 94. Suggested "cleared by the immune system".

Line 95. Extra space before period.

Line 96. Finish the sentence, instead of "this" use "immune evasion" or "immune clearance" etc.

Line 109. I found the sentence convoluted: "VSG genes are transcribed by RNA Polymerase I at an extra nucleolar transcription factory termed the expression site body (ESB)(23, 24) from a subtelomeric locus, called the expression site (VSG-ES)." I also suggest replacing "transcription factory" to "extranucleolar foci" since the function of ESB remains unclear. Rewrite the sentence, please.

Line 112. The reference 25 is wrong. I suggest Becker M et al. 2004 Genome Res.

Line 113. The sentence seems wrong: "and encode of several proteins coding genes". Check English.

Line 137: "high rates of Pol 1 transcription are facilitated by is facilitated by the deposition of TDP1" Check duplication of words.

Line 361: Fig legend: "During antigenic variation donor VSGs are from a silent VSG-ES, minichromosome, or subtelomeric VSG array are used." Check the structure of this sentence.

Line 164. "This suggests that trypanosomes must ensure that the time to transmission is less than the time to VSG repertoire exhaustion." It may be better to rewrite this sentence as trypanosomes do not "ensure". Transmission may occur before VSG repertoire exhaust, and exhaustion of VSG gene repertoire may not occur due to recombination potential.

Line 179. "Components of the inositol phosphatase pathway, here phosphatidylinositol 5-kinase - TbPIP5Pase and phosphatidylinositol 5-phosphatase - TbPIP5K, modulate switch frequency through interactions with telomeres (53)." It should be inositol phosphate pathway instead of inositol phosphatase pathway. Also, the abbreviations for PIP5K and PIP5Pase seems inverted. PIP5K is the kinase and PIP5Pase is the phosphatase.

Line 181. "Indeed, TbPIP5Pase has been shown to interact with TbRAP1 and TbTRF (52)." The reference provided is wrong (52, Jehi SE et al. NAR 2014). Jehi did not show interactions of PIP5Pase with RAP1. The correct reference should be Cestari and Stuart PNAS 2015 and Cestari et al. 2019 Mol Cell Biol. Also, only RAP1 was found to interact with PIP5Pase, not TRF. This was detailed in Cestari et al. 2019 Mol Biol Cell.

Line 191. "most common used either". It seems that a verb is needed in this sentence.

Line 192. "Selectable markers enable VSG-ES to be tagged with either positive or negative selectable markers". The sentence is cumbersome; please review it. Also, explain better the process. Line 194 says "with drug treatment," which drug, and how does it work?

Line 203. "results cells that have switched ins order to survive" Misspelled words.

Line 216. "similar to that seen yeast mating type switching (69), but is unlikely give the trypanosomes ability to undergo gene conversion in the absence of the 70bp repeats". There are several small grammatical issues in this text which causes a tremendous distraction. From now on I will stop commenting on them and strongly recommend the authors to review the English.

Line 234-235. Please clarify in the text the hierarchy in VSG switching. Does the order or preference in ESs and minichromosomes for VSG switching the consequences of preferred DNA sequences in these sites that may facilitate recombination? If so, are they not appearing so often at later stages of infection because the host has developed antibodies for VSGs in these more favourable recombination sites?

Line 245. I suggest using "the advantage" instead of "The purpose", because this is the result of an evolutionary process.

Line 267. BRCA2 and BRC repeats. Please, state what they stand for. Same to other abbreviations.

Line 269. Reference 69 is a work in yeast and not in trypanosomes. The text mentions the role of TbTOPO3a in ES recombination using 70 bp repeats. Please correct the reference.

Line 292. "The requirements for HR go beyond the mere genetic factors, we are now discovering that" confusing sentence with grammatical mistakes.

Line 318. Review your grammar. Do this to all the text.

Author's Response to Decision Letter for (RSOB-19-0182.R0)

See Appendix A.

Decision letter (RSOB-19-0182.R1)

21-Oct-2019

Dear Dr Glover,

We are pleased to inform you that your manuscript entitled "Escaping the immune system by DNA repair and recombination in African trypanosomes" has been accepted by the Editor for publication in Open Biology.

Sincerely,

The Open Biology Team
mailto:openbiology@royalsociety.org

Appendix A

Editors Comments to Author:

We are pleased to inform you that your manuscript RSOB-19-0182 entitled "Escaping the immune system by DNA repair, recombination in African trypanosomes" has been accepted by the Editor for publication in Open Biology. The reviewer(s) have recommended publication, but also suggest some minor revisions to your manuscript. Therefore, we invite you to respond to the reviewer(s)' comments and revise your manuscript.

We are pleased to hear that our manuscript has been accepted by the editor for publication. We are also grateful to both reviewers for their constructive comments and suggestions. We have made all the changes requested by the reviews and believe that this manuscript is substantially improved. The relevant changes are highlighted in blue in the revised manuscript and the new line numbers given where relevant.

Yours sincerely,

Lucy Glover

Reviewer(s)' Comments to Author:

Referee: 1

Comments to the Author(s)

Teruel et al provide a timely review of the importance of DNA repair and recombination to antigenic variation in *Trypanosoma brucei*. Their review complements recent reviews by Aresta-Branco et al (parallels with antibody diversification; PMID: 30826207), da Silva et al (mechanisms of DNA lesion generation; PMID: 30440029), Bangs (evolution of antigenic variation; PMID: 30370931), and Cestari and Stuart (transcriptional regulation; PMID: 29491740).

The review is well organised, clearly presented and addresses important issues of relevance to those interested in antigenic variation, as well as the wider area of DNA repair. However, there are a few issues that should be addressed prior to publication, as well as a number of typos (see list after comments).

Major comments

1. The authors provide a detailed summary of switch rate variation seen between strains and using different approaches (lines 190-208). How might these variations influence or complicate the authors' analyses and those of others in the field? Is there a risk that the culture adaptation effect (line 207) biases the types of DNA repair (and breaks) that drive antigenic variation? Is this issue relevant to investigation of the apparent switching mechanism hierarchy (line 166, 235, 324, 328)?

We have now added the following paragraph to the text: 'While these data are instructive, they rely on experimental systems which are far from ideal. The strains used are largely culture adapted monomorphic strains, which seem to display greatly reduced switching frequencies. The experimental limitations are, however understandable given the difficulty of the task. The ideal situation would be to measure the switching frequency in a chronic model infection, however this is not possible using current technology. The predominant form of switching in trypanosomes is via DNA recombination, yet the majority of the above-mentioned techniques measure *in situ* switching frequencies, or indeed, in the case of VSG RNAi, result in *in situ* switchers. It remains to be seen whether the current data on VSG switching frequencies accurately reflect long-term trypanosome infections. Lines 214 - 223

2. The description of the role of HAT3 in repair and VSG switching is confusing (lines 298-303) – as written, the second sentence doesn't logically follow the first. The authors should include more detail, emphasising the different effects of HAT3 at internal chromosomal loci versus VSG expression sites; hence HAT3's suppressive effect on VSG switching (Glover & Horn 2014; PMID: 25300492).

We have changed this section to now read: ' HAT3's function in DNA repair was shown to be locus specific and impact VSG switching; at a chromosome internal locus, HAT3 promoted DNA resection and RAD51 focal assembly but suppressed resection at a VSG-ES. The consequence of this is that in the presence of HAT3 VSG gene conversion events, which required resection to expose the 70-bp repeats and provide the homology required for repair, are suppressed (92) (Figure 2). This suggests that specific chromatin marks may regulate DNA recombination and VSG switching. Lines 311 - 316

Minor comments

3. Some of the word choices have unintended implications. For example, 'cyclical' (line 49) suggests a return to a previous state, whereas antigenic switching and clearance of specific antigenic types by the adaptive immune system results in the progressive expansion and clearance of new variants, i.e. a process of ongoing change enabling persistence. Also, 'several orders of magnitude' (line 53) seems an exaggeration, given the numbers provided at lines 56-8.

We have changed 'cyclical' for 'continual' Line 49

We have changed 'several' to 'between one and two' Line 52

4. I noted some inconsistencies, e.g. 'one third' (line 72) contradicts the text at line 52 (10%), and ESAGs don't directly contribute to the variant antigen repertoire as this sentence implies.

We have changed the text at Line 71 to 20 % and removed the text at line 52.

The statement at line 259-61 seems inconsistent with that at line 143-4 – 'switching primarily driven by gene conversion' (143), 'recq2 mutants show elevated switching by [...] gene conversion. [...] a marked shift in the type of switching used [...] compared to wild type cells'. The authors should also review for repetitive text (e.g. at lines 283-4).

We have changed the sentence at line 272-4 to 'Two RecQ-like helicases have been identified in trypanosomes and *recq2* mutants show elevated VSG switching by telomere recombination in addition to VSG gene conversion events (78).'

We have also removed the repetitive text. Line 296

5. Rather than using the term 'increasing virulence' (line 105), the authors should be more specific and say that the addition of O-glycosylation to the already extensive VSG variability potentially extends the ability of the parasite to evade the host's adaptive immune system.

We have changed this to 'increasing the trypanosomes ability to escape the hosts adaptive immune system'. Line 106-107

The authors should also be more precise in their description of the position of the most productive DSBs in the active VSG-ES (line 233). My understanding is that this is thought to be limited to an upstream region proximal to the VSG, probably in or close to the 70 bp repeats (rather than further upstream in the ESAGs or towards the distal upstream ES promoter).

We have changed this sentence to 'breaks formed between the VSG gene and the 70 bp repeats'. Line 244-245

6. The description of the RNAi experiments, the emergence of switch variants, and the impact of culture adaptation on switch frequency (lines 202-8) needs rewriting for clarity. In particular, the relevance of the essential nature of VSG in this context is unclear and it should be stated that the actively expressed VSG is specifically targeted, enabling a switch to a new, non-targeted variant.

We have changed this sentence to now read: 'In trypanosomes, because VSG expression is essential, RNAi against the active VSG transcript results in cells that have switched to VSGs that are not targeted by RNAi. The switching frequency following VSG RNAi is estimated to be at a rate of 10^{-4} per division, predominantly by *in situ* switching.' Line 207-210

Also, I was unsure as to the relevance of VSG-associated growth rates to this discussion (lines 208-11).

We have removed this final sentence.

7. The numbers of mini and intermediate chromosomes are less constrained than stated at line 108. There are ~100 minichromosomes and intermediate chromosome ploidy varies between strains (see Wickstead et al 2004; PMID: 15173109).

We have corrected this point. Line 111

8. At line 276, the authors state there are five RAD51-related proteins but list four, omitting DMC1 (Proudfoot and McCulloch 2005; PMID: 16326865).

We have corrected this point. Line 289

9. As the focus of the review is DNA repair, I suggest the authors re-order their final questions to give primacy to those pertinent to this topic in their list starting at line 317.

We have re-ordered this as suggested, starting line 333

Typos and suggested edits

We have made all the corrections listed here.

Line 56 ('...families number, for example, ~60 var genes'); 112 (state that VSG expression sites are found at the sub-telomeres of megabase chromosomes); 121 (delete 'they'); 141 (delete second 'is facilitated by'); 135, 138 ('cohesin'); suppress ('untranslated'); 181 ('suppress'); 184-5 (put brackets around abbreviations); 190 ('preserve'); 196 ('uses', 'assesses'); 199 ('in situ switching frequencies are estimated at'); 228 ('seen in yeast', 'given'); 257 ('preserves'); 268 (italicise 'Leishmania'); 269 ('hypersensitive to DNA'); 273 (consistent terminology for null mutants); 274 (delete 'that'); 280 ('via an expanded'); 283 ('DNA damaging agents'); 290 ('RAD51 in response to DNA damage'); 295 ('suppresses'); 305 (delete 'a'); 339-341 ('does the trypanosome gain by'; 'VSG genes expressed in these'); 342 ('antigenicity expand the parasite's ability'); 337 ('Given that there is a loose hierarchy of VSG switching, does repair pathway choice change over time and, if so, how is this regulated?'); 370 ('single megabase chromosome'); 371 ('found proximal to the telomere'); 373 ('light grey boxes'); 374 ('variation, donor VSGs from'); 379 (delete 'highlighted in black'); 384 ('MRN complex (light blue circle) binds and initiates resection')..

Referee: 2

Comments to the Author(s)

This is a helpful review by Teruel and colleagues. The work submitted here provides a good overview with some discussions (not detailed) on the mechanisms underlying antigenic variation in African trypanosomes focusing on recombination. The work covers well the literature and discusses many possible mechanisms revolving VSG recombination and potential mechanisms that initiate the switching process. It would be nice to have this potential mechanism that initiates the switching numbered to highlight this part of the discussion since a non-expect may not grasp the relevance of the content here.

There are minor faults on the content which can be easily fixed, and I have provided some comments that will help the authors. This manuscript has a vast number of grammatical mistakes, some of which is pointed out here (but not all). Hence, I suggest the authors review their manuscript carefully.

Detailed comments:

line 51 - A massive expansion in VSG genes have resulted in ... Is this meant to be a massive expansion of VSG genes that resulted in ...?

We have changed this sentence. Line 50-52

line 72: the sentence " and their associated sequences..." is unclear. Is this meant to be VSG related sequences or VSG pseudogenes

We have deleted 'and their associated sequences' and change the sentence at Line 72

line 82: "as a trigger", it is better to complete the sentence: a trigger for switching if that is what you mean.

We have added 'for VSG switching' line 82

Line 88: " VSGs are attached to the cell by a glycosylphosphatidylinositol (GPI) anchor (14), which protects the cell from complement mediated complement mediated lysis (15) and shields invariant surface molecules also present on the surface." This sentence is unclear because it indicates that GPIs confer resistance to complement. GPIs anchors the VSG protein on the surface.

We have added ' forming a coat' to this sentence. Line 90

Also, the reference 15 (Sheader K et al. PNAS 2005) do not show the involvement of VSGs and complement resistance. It shows that knockdown of VSG gene results in clearance of the parasite during infection of mouse. The role of VSG in complement resistance was surprisingly superficially studied in this field and here some helpful references that the author can start his/her search: Engstler M et al. Cell 2007; Russo DC et al 1994 Parasitol Res. This is not to say that complement will not have a role, but it appears that other mechanisms of clearance are also involved (e.g., J A Macaskill et al Immunol 1980; Cheung JL et al 2016 Plos Pathogen).

We have changed the references here as suggested. Line 90

Line 94. Suggested "cleared by the immune system".

We have changed this as suggested. Line 95

Line 95. Extra space before period.

We have removed the extra space. Line 97

Line 96. Finish the sentence, instead of “this” use “immune evasion” or “immune clearance” etc.

We have added ‘immune evasion’. Line 98

Line 109. I found the sentence convoluted: “VSG genes are transcribed by RNA Polymerase I at an extra nucleolar transcription factory termed the expression site body (ESB)(23, 24) from a subtelomeric locus, called the expression site (VSG-ES).” I also suggest replacing “transcription factory” to “extranucleolar foci” since the function of ESB remains unclear. Rewrite the sentence, please.

We have rewritten this sentence. Lines 111-115

Line 112. The reference 25 is wrong. I suggest Becker M et al. 2004 Genome Res.

We have changed this reference.

Line 113. The sentence seems wrong: “and encode of several proteins coding genes”. Check English.

We have corrected this sentence. Lines 116

Line 137: “high rates of Pol 1 transcription are facilitated by is facilitated by the deposition of TDP1” Check duplication of words.

We have deleted the duplicated words. Line 141

Line 361: Fig legend: “During antigenic variation donor VSGs are from a silent VSG-ES, minichromosome, or subtelomeric VSG array are used.” Check the structure of this sentence.

We have changed this sentence to ‘During antigenic variation donor VSGs from silent VSG-ES, minichromosomes, or subtelomeric VSG arrays are used’. Line 374-5

Line 164. “This suggests that trypanosomes must ensure that the time to transmission is less than the time to VSG repertoire exhaustion.” It may be better to rewrite this sentence as trypanosomes do not “ensure”. Transmission may occur before VSG repertoire exhaust, and exhaustion of VSG gene repertoire may not occur due to recombination potential.

We have adjusted this sentence. Line 168

Line 179. “Components of the inositol phosphatase pathway, here phosphatidylinositol 5-kinase - TbPIP5Pase and phosphatidylinositol 5-phosphatase - TbPIP5K, modulate switch frequency through interactions with telomeres (53).” It should be inositol phosphate pathway instead of inositol phosphatase pathway. Also, the abbreviations for PIP5K and PIP5Pase seems inverted. PIP5K is the kinase and PIP5Pase is the phosphatase.

We have corrected this sentence. Line 184-185

Line 181. “Indeed, TbPIP5Pase has been shown to interact with TbRAP1 and TbTRF (52).” The reference provided is wrong (52, Jehi SE et al. NAR 2014). Jehi did not show interactions of PIP5Pase with RAP1. The correct reference should be Cestari and Stuart PNAS 2015 and Cestari et. al 2019 Mol Cell Biol. Also, only RAP1 was found to interact with PIP5Pase, not TRF. This was detailed in Cestari et al. 2019 Mol Biol Cell.

We have corrected these references. Line 185-6

Line 191. “most common used either”. It seems that a verb is needed in this sentence.

We have added ‘are’ into this sentence. Line 196

Line 192. “Selectable markers enable VSG-ES can to be tagged with either positive or negative selectable markers”. The sentence is cumbersome; please review it. Also, explain better the process. Line 194 says “with drug treatment,” which drug, and how does it work?

We have changed this sentence to ‘VSG-ES can be tagged with either positive or negative selectable markers, and the frequency of parasite survival analysed following antibiotic selection. Using positive selectable markers, *in situ* switching frequencies are estimated at between 1×10^{-4} and 6×10^{-7} , depending on the VSG-ES (64, 65)’. Line 198-199

Line 203. "results cells that have switched ins order to survive" Misspelled words.

We have corrected this sentence. Line 208

Line 216. "similar to that seen yeast mating type switching (69), but is unlikely give the trypanosomes ability to undergo gene conversion in the absence of the 70bp repeats". There are several small grammatical issues in this text which causes a tremendous distraction. From now on I will stop commenting on them and strongly recommend the authors to review the English.

We have corrected this sentence. Line 228-229

Line 234-235. Please clarify in the text the hierarchy in VSG switching. Does the order or preference in ESs and minichromosomes for VSG switching the consequences of preferred DNA sequences in these sites that may facilitate recombination? If so, are they not appearing so often at later stages of infection because the host has developed antibodies for VSGs in these more favourable recombination sites?

We have adjusted this sentence and added a statement to clarify the text. Line 246-250

Line 245. I suggest using "the advantage" instead of "The purpose", because this is the result of an evolutionary process.

We have changed this as suggested. Line 259

Line 267. BRCA2 and BRC repeats. Please, state what they stand for. Same to other abbreviations.

We have done this where possible.

Line 269. Reference 69 is a work in yeast and not in trypanosomes. The text mentions the role of TbTOPO3a in ES recombination using 70 bp repeats. Please correct the reference.

We have corrected this reference. Line 301

Line 292. "The requirements for HR go beyond the mere genetic factors, we are now discovering that" confusing sentence with grammatical mistakes.

We have changed the text to read: 'We are now discovering that HR requires both genetic factors and PTMs of histones and non-histone proteins for the initiation and execution of the repair response' line 303-304

Line 318. Review your grammar. Do this to all the text.

We have reviewed the text as suggested for grammatical errors.